# Assessment of Carbon Productivity Dynamics in Aspen Stands under Climate Change Based on Forest Inventories in Central Siberia

Andrey Andreevich Vais, Valentina Valerievna Popova, Alina Andreevna Andronova, Viktor Nikolaevich Nemich, Artem Gennadievich Nepovinnykh and Pavel Vladimirovich Mikhaylov *

Scientific laboratory "Forest Ecosystems", Reshetnev Siberian State University of Science and Technology, 660037 Krasnoyarsk, Russia
* Correspondence: mihaylov.p.v@mail.ru

**Abstract:** The aim of the present research was to study the dynamics of growth and conditions of aspen stands under climate change, according to different periods of forest inventory. The study was conducted in modal aspen forests growing in the subtaiga/forest steppe region of Central Siberia. Aspen forests grow intensively at young age, which allows them to realize maximum carbon sequestration potential. The research was based on forest inventory data from 1972, 1982, 2002, and 2021 (the study was conducted on a limited territory). There was a steady increase in temperatures in the growing season from 1982 to 2002. The amount of precipitation in the same season and period, however, did not exceed the median value. With an increase in the sum of temperatures in 1982–2002 from 1800 °C to 2100 °C, carbon stored in the stands increased from 0.56 to 1.48 tons C/ha per year. This statement is true for pure aspen forests aged from 10 to 30 years. There is a certain (although indirect) influence that climate trends have on aspen forests' carbon dynamics. There was a decrease in the average carbon increment in aspen forests from the age of 40. After 55 years, the average carbon increment values in the aspen forests leveled off, and the differences depending on the stand composition became insignificant. Along with an increase in biomass increment with age, aspen stands started losing resilience, and trees began to die due to natural and pathogenic mortality. At ages between 50 and 80, carbon emission increased from 1 to 12 tons C/ha.

**Keywords:** aspen; climate; stock; carbon productivity; mean annual increment

## 1. Introduction

Climate change creates effects on forest ecosystems globally [1]. There is a whole range of factors that influence long-term changes in forest ecosystems: features of biocenosis and natural processes of its development, climatic conditions, and anthropogenic, pyrogenic, and biogenic impacts.

Climate change poses a number of threats to boreal forests, such as biodiversity loss, an increase in wildfire frequency and burned areas, and insect/pest outbreaks [2]. Scientists believe that the influences of climate change indicators (carbon dioxide concentrations, temperature fluctuations, precipitation variability) could have significant implications on the carbon sequestration potential of forest ecosystems [3–5]. It is likely that the role of boreal forests as carbon sinks will decline [6]. There are models predicting that carbon sink in the forests of Russia will decrease, even at a low level of active forest management [7].

Forest carbon projects are considered to be one of the key methods to reduce greenhouse gas emissions and slow global warming. The forest carbon projects are aimed at speeding up the rate of carbon sequestration in forests by implementing the following measures: conserving intact forests, preventing forest fires, and limiting other factors causing forest destruction, intensive forestry, reforestation, and afforestation; innovative logging

technologies, converting non-forest land to forests, multi-purpose forest management, and converting timber into wood products [6–8].

What is more, some researchers believe [9,10] that forest climate projects can be profitable.

Selecting tree species that present higher values of carbon storage can play a crucial role in forest climate projects.

The Eurasian aspen (*Populus tremula* L., 1753) is one of the fastest growing tree species in Siberia [11]. This is a pioneer species that actively colonizes open sites (clearcuts, burnt areas, glades, etc.). The high growth rate and biomass production contribute to the significant carbon sequestration of aspen forests. We believe that there is a need to study the possibility of using aspen in forest carbon projects and the subsequent creation of carbon farms, including the research focused on growth dynamics, carbon pools, and sustainability in aspen forests.

To assess the effectiveness of climate change projects, the volume of additionally sequestered carbon is compared to the natural trend. However, there are certain issues in describing natural trends (baseline) and justifying and creating a project scenario. To solve these issues, data on growth dynamics and the state of forests growing in a given area is needed.

Despite the large number of studies [3,4,6,12,13] focused on carbon regulation in Russia, the task of creating a database of carbon content in ecosystems has not lost its relevance.

Notably, studies of long-term changes in deciduous species in Siberia were carried out on a limited and fragmented basis on individual research plots.

The approaches available in Russia make it possible to estimate carbon sequestration for large areas using statistical reports data, including forest inventories [12]. The methodology is applicable at macro-regional scale (for example, for Central Siberia). However, at a local scale, the methodology bears significant errors.

The study of growth dynamics and the state of forests is based on a quantitative assessment of forest inventory indicators by age periods [14–17].

Global and regional climate changes are assumed to affect carbon productivity and the carbon sequestration potential of forest ecosystems. Therefore these changes should be reflected in forest inventories for a 50-year period (1972–2021).

Forest phytomass carbon stock dynamics under global climate change are estimated by two approaches (empirical and mechanistic models) based on regression analysis and object description [13].

The purpose of the present study was to assess the productivity of aspen forests by the average increments of woody biomass and carbon in them and considering stand composition, age, and climate change.

## 2. Materials and Methods

### 2.1. Study Area

The study was carried out in modal aspen stands growing near Krasnoyarsk in the educational and experimental forest unit of the Reshetnev Siberian State University of Science and Technology (Russia) (Figure 1).

Intensive initial growth is typical of aspen forests, which allows them to realize carbon sequestration potential to the maximum extent. The research was based on the forest inventory conducted at the following periods: 1972 (428 forest units), 1982 (305 forest units), 2002 (454 forest units), and 2021 (21 forest units in 6 forest compartments). We selected aspen-dominated forest compartments (pure and mixed stands). The total number of forest units was 1208 (the total number for all inventory periods). All plots, regardless of the study period, were characterized by a standard set of taxation indicators, including age, average height, average diameter, stock, *bonitet* class, relative density, forest type, merchantability class.

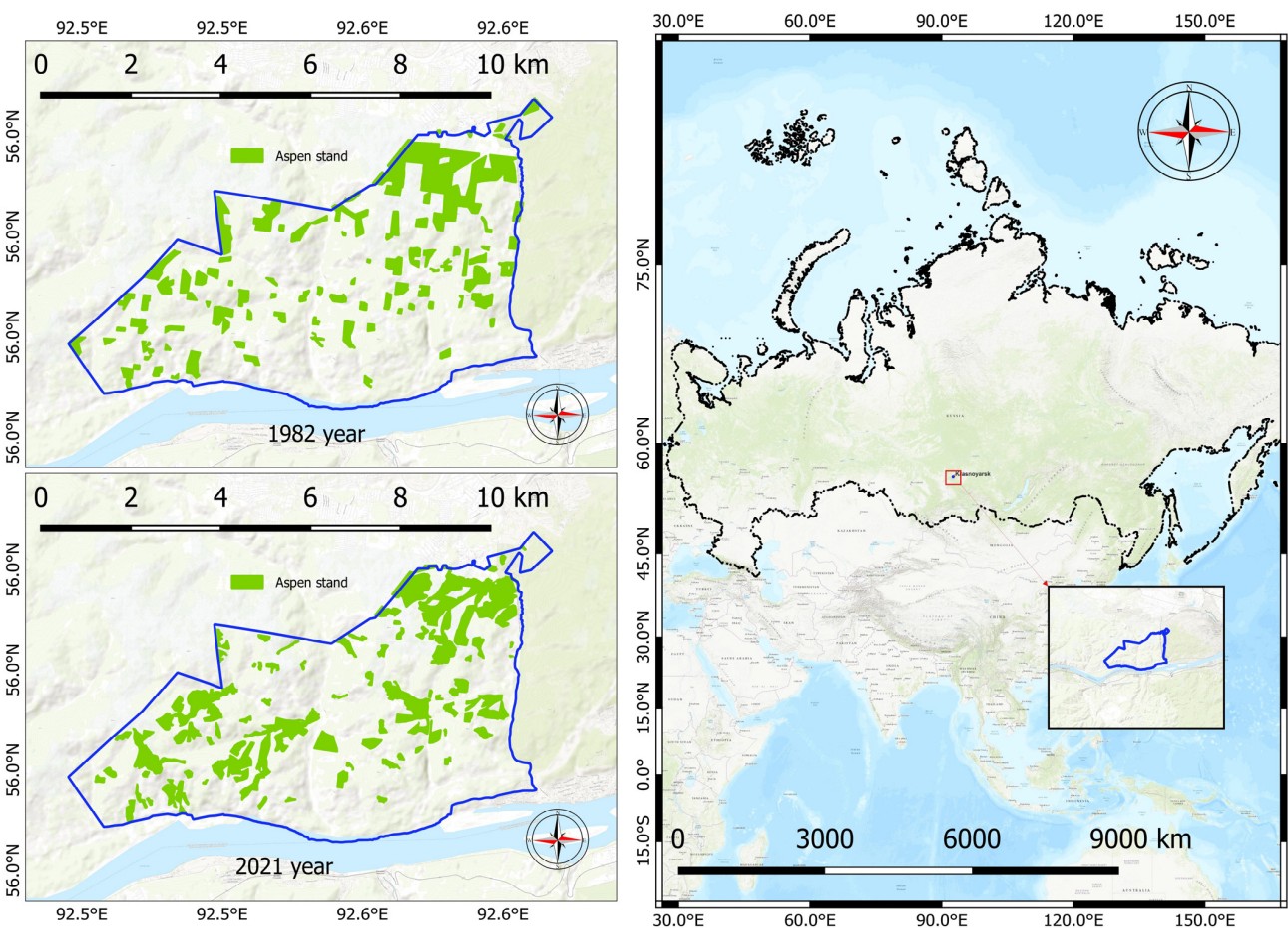

**Figure 1.** Study area outline map (right side) and dynamics of the area covered by the aspen forests according to the data of 1982–2021.

## 2.2. Data Collection Methods

The balance approach applied in ecology and forestry was used as a methodological basis for calculating the forest carbon budget. The method is based on age-related changes in carbon stock. However, the average increment rate may differ significantly from the current stock change. For example, the average carbon stock increment is much lower than the current increase in young stands, but higher in mature and overmature stands. Nevertheless, the method is considered to provide acceptable results, since the noted discrepancies have a different sign and cancel each other out [18]. Experience shows that there is no method offering accurate results for estimating the current stock change. Methodology proposed by the All-Russian Research Institute for Silviculture and Mechanization of Forestry [19] involves calculation of annual carbon budget based on average carbon increment. The annual increase in above-ground tree carbon stocks is obtained by multiplying the average annual increase in phytomass for each tree species by the occupied area and summing up for forestry, the federal subjects of Russia, and boreal zones as a whole [20].

A field study was carried out in 2021 in the form of a detailed pathological inspection following generally accepted methods [21,22] including preliminary and detailed surveys. During the preliminary survey, a visual assessment of aspen forests condition and health was made within the compartments of at least 1.5 ha. The detailed survey was carried out on 14 research plots (RP) placed in typical tallgrass aspen forests. We assessed 130–180 trees on each research plot, and divided them into four-centimeter diameter classes and condition classes: 1—with no signs of weakening; 2—weakened; 3—severely weakened; 4—dying; 5—dead (lost viability): current and previous-years snag, windthrow, and windsnap.

Tree condition class was determined mainly by the condition of tree crowns. We also recorded trees affected by diseases, which were determined by a complex of specific and indirect signs.

Preliminary analysis of the initial data (sorting and diagramming) was performed in the Microsoft Excel spreadsheet by Microsoft.

To process and analyze data (samples) obtained during the research we used statistical analysis (STATISTICA 10) and regression analysis (Curve Expert). The accepted level of significance was $p \leq 0.05$. Statistical calculations were carried out using STATISTICA 10 and Curve Expert programs.

*2.3. Data Analysis*

Graphical and analytical approaches were used to conduct the data analysis.

The average increment in stock was determined by the standard formula:

$$\Delta M = M/A, \tag{1}$$

In Equation (1), $\Delta M$ is the average increment in stock, m³ ha⁻¹ year⁻¹, $M$ is the growing stock, m³/ha; $A$ is the age in years.

The average increment in stock depends on the factors that determine the stock volume (*bonitet* class, density, composition, average height, accuracy of stock evaluation in young stands). The age of a forest stand also influences growth process. Studied aspen forests were modal, their density varied from 0.4 to 0.8, rarely 0.9; *bonitet* class II–III; aspen took from 40 to 100% in stands' composition. A wide range of the stands' parameters explained the high variability in average growth values. Such differentiation is important when assessing the forests response to temporal climate trends. The average increment values were differentiated (sorted) by composition coefficients (40%–100% in a stand composition) and five-year periods. Then, we calculated the main statistics using descriptive statistics in STATISTICA 10 software at a confidence level of 95.4%. In all cases, the initial samples of 1972, 1982, and 2002 were representative (experimental accuracy $p < 5.0\%$), and the average growth parameters were significant according to Student's test (tf > ttab) at a significance level of $p = 0.05$.

We used the following data from the NASA POWER portal (https://power.larc.nasa.gov/docs/methodology/data/sources/ (accessed on 15 October 2022) to study climate trends: temperature at 2 m height, °C; amount of precipitation, mm/day. Meteorological parameters were derived from the NASA's GMAO MERRA-2 assimilation model and GEOS 5.12.4 FP-IT. Spatial resolution: 0.5° × 0.625°.

The sum of temperatures above 5 °C and the precipitation sum for the studied period (1982–2020) were summarized within the growing season. Then, we analyzed and compared these data with a median value for the fifty-year period. We added the following lines on all graphs (except for the actually observed indicator by year) to represent the trend: the median lines of the indicator for the entire observation period (red horizontal line) and a LOESS-type smoothing line with a confidence interval (blue line).

The carbon volume was determined by the average increment in stock. To do this, we used the formula recommended by the Intergovernmental Panel on Climate Change in 2006 [23]:

$$Gw = Iv \times D \times BEF \times (1 + R) \times CF, \tag{2}$$

In Equation (2), *Gw*—average annual aboveground biomass carbon increment, tons C/ha per year; *Iv*—average increment in stem stock volume, m³/ha per year; *D*—basic wood density, tons dry matter/m³ merchantable volume (for different tree species varying from 0.3 to 0.6 tons d. m./m³ stem volume), 0.510; *BEF*—biomass expansion factor for conversion of merchantable volume to aboveground tree biomass; *R*—root-to-shoot ratio (for different tree species varies from 0.2 to 0.3); *CF*—carbon fraction of dry matter (default = 0.5), tons C/tons d.m.)

## 3. Results

Firstly, we determined the extent to which aspen forests' composition influences the average growth value. The initial data was sorted by the share of aspen in a stand composition. Then we calculated the average stock increment depending on the coefficient of aspen share in a stand composition. Histograms (Figure 2) show the results for the analysis of 1972–2002 forest inventory data. We chose this period to study due to the intensive growth of aspen stands.

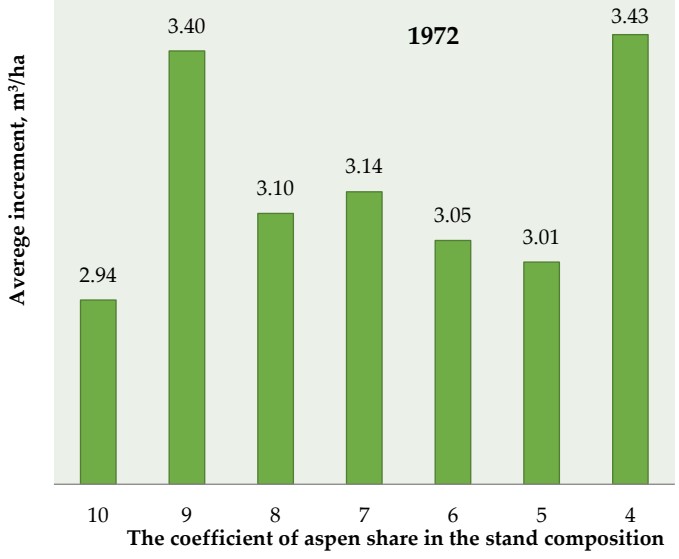
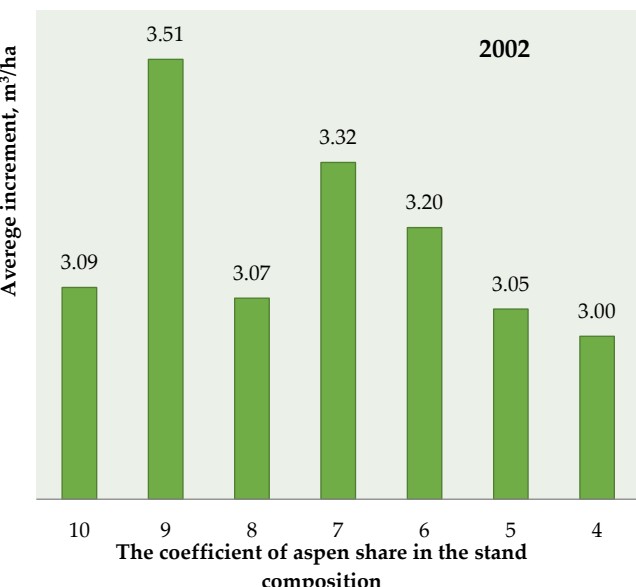

**Figure 2.** Correlation between the average stock increment and the coefficient of aspen share in the stand composition (1972, 2002).

At the next stage, the task was to reduce the variability of the average increment value. To do this, we differentiated the initial data by age in five-year increments. Table 1 shows regression models that reflect the relationship between the average stock increment and aspen forests' age. We selected the most adequate rational function that fitted the following conditions: maximum correlation coefficient between the experimental and leveled data (R), minimum standard error (mx), and the significance of the regression equation coefficients.

**Table 1.** Parameters of regression models identifying the relationship between the average stock increment and the age of aspen forests.

| Relationship | Equation | Function Coefficients | | | | R | $m_x$ |
|---|---|---|---|---|---|---|---|
| | | a | b | c | d | | |
| $\Delta c = f(A)$ 1972 | Rational Function $\Delta c = (a + b \times A)/$ $(1 + c \times A + d \times A^2)$ | $-3.37 \times 10^{-2}$ | $7.45 \times 10^{-2}$ | $-2.29 \times 10^{-2}$ | $5.17 \times 10^{-4}$ | 0.94 | 0.41 |
| $\Delta c = f(A)$ 1982 | | $-2.79 \times 10^{-3}$ | $5.81 \times 10^{-1}$ | $8.23 \times 10^{-2}$ | $1.28 \times 10^{-3}$ | 0.98 | 0.30 |
| $\Delta c = f(A)$ 2002 pure | | $-3.63 \times 10^{-3}$ | $1.38 \times 10^{-1}$ | $-2.14 \times 10^{-2}$ | $7.40 \times 10^{-4}$ | 0.95 | 0.52 |

Note: $\Delta c$—average current increment in stock, $m^3/ha$; A—age, years; pure—pure aspen forests; a, b, c, d—rational function coefficients; R—the correlation coefficient between the experimental and leveled data; $m_x$—standard error, $m^3/ha$. All coefficients of the equation were statistically significant ($p < 0.05$). Regression estimates were obtained at a confidence level of 0.954.

The main factors influencing the growth and development of plantations are the average annual temperature, the difference between the average July and January temperatures, and the average annual precipitation [3–5]. In this regard, we constructed diagrams reflect-

ing the dynamics of the most significant climatic indicators over the years (Figures 3 and 4).

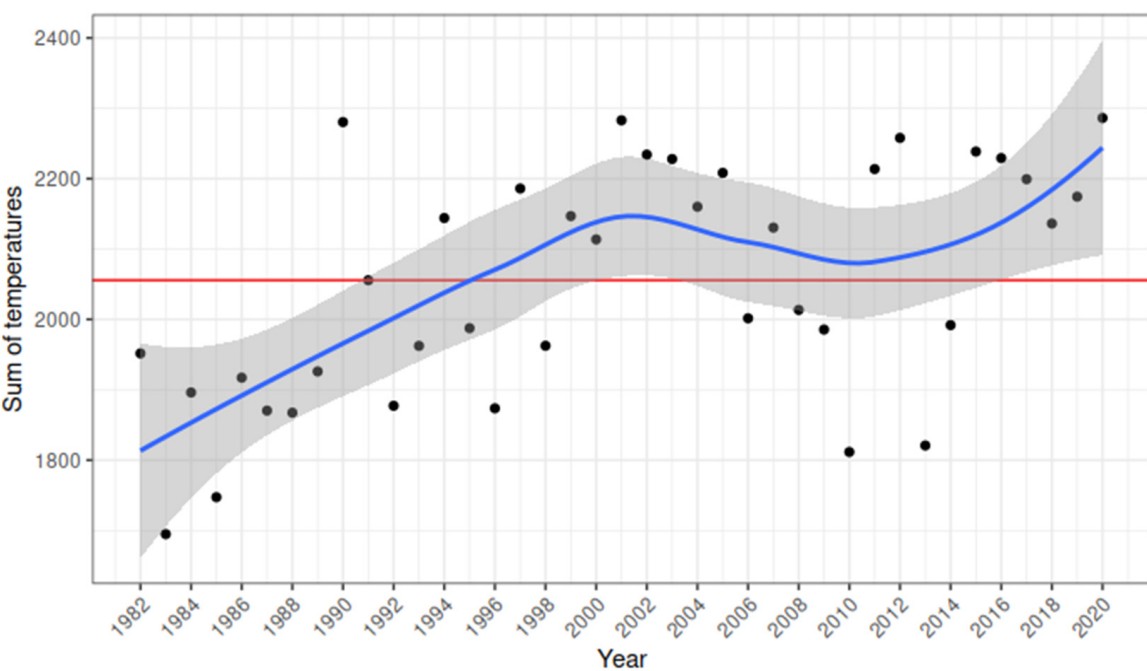

**Figure 3.** Relationship between the sum of temperatures above five degrees (t > +5 °C) and the year of observation.

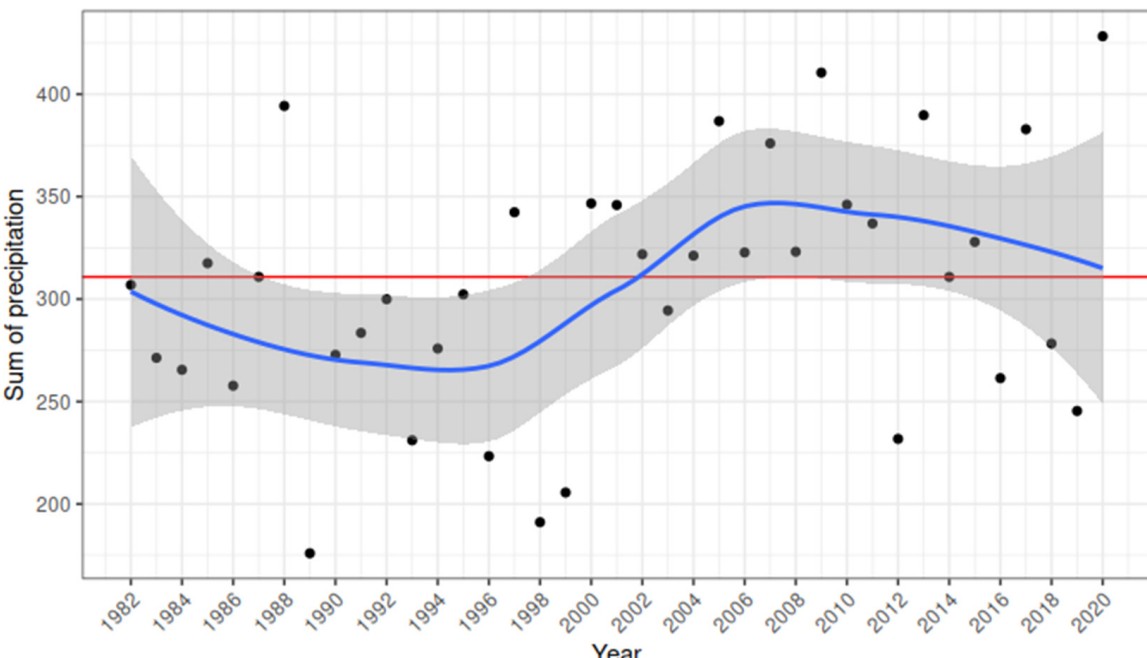

**Figure 4.** The relationship between total precipitation during the growing season and the year of observation.

Based on the equations obtained (Table 1), we calculated the adjusted values of the average stock increment for the presented growth periods according to the forest inventory of 1972, 1982, and 2002. We can state the following: differences in growth were expressed at a young age; by the age of 50, the values of the average growth in aspen forests did not differ significantly. The forest inventories of 1982 and 2002 proved the influence of a stand composition (pure or mixed stands) on the average growth rate.

We revealed an impact of climate trends; however, this is indirect. The minimum growth values were observed in 1972 (pure and mixed aspen forests), as well as in pure stands in 1982. The maximum average increment was observed in pure aspen stands in 2002. The sum of positive temperatures during the growing season had a positive effect on the average growth rate in young pure aspen forests. Nevertheless, the role of a stand composition (pure or mixed) exceeded the role of climate change in average growth dynamics.

Table 2 presents the calculations of carbon in the average growth according to Formula (1).

**Table 2.** Average annual carbon stock increment in living biomass (tons C/ha per year) by aspen forests' age and inventory periods.

| Age, Years | 1972 | | 1982 | | 2002 Mixed Stands | | 2002 Pure Stands | |
|---|---|---|---|---|---|---|---|---|
| | m³/ha | Increment, Tons C/Year | m³/ha | Increment, Tons C/Year | m³/ha | Increment, Tons C/year | m³/ha | Increment, Tons C/Year |
| 5 | 1.40 | 0.45 | 2.01 | 0.65 | 0.70 | 0.23 | 0.71 | 0.23 |
| 10 | 1.94 | 0.63 | 2.97 | 0.96 | 1.38 | 0.45 | 1.73 | 0.56 |
| 15 | 2.44 | 0.79 | 3.45 | 1.11 | - | - | - | - |
| 20 | 2.83 | 0.91 | 3.68 | 1.19 | 2.64 | 0.85 | - | - |
| 25 | 3.09 | 1.00 | 3.75 | 1.21 | - | - | 4.30 | 1.39 |
| 30 | 3.23 | 1.04 | 3.77 | 1.22 | 3.41 | 1.10 | 4.59 | 1.48 |
| 35 | 3.26 | 1.05 | 3.73 | 1.20 | 3.58 | 1.16 | - | - |
| 40 | 3.21 | 1.04 | 3.67 | 1.18 | 3.63 | 1.17 | 4.36 | 1.41 |
| 45 | 3.11 | 1.00 | 3.58 | 1.16 | 3.60 | 1.16 | 3.99 | 1.29 |
| 50 | 2.98 | 0.96 | 3.49 | 1.13 | 3.51 | 1.13 | 3.61 | 1.17 |
| 55 | 2.83 | 0.91 | 3.40 | 1.10 | 3.39 | 1.09 | 3.29 | 1.06 |
| 60 | 2.68 | 0.87 | 3.31 | 1.07 | 3.25 | 1.05 | 3.10 | 1.00 |
| 65 | - | - | 3.21 | 1.04 | 3.10 | 1.00 | 3.07 | 0.99 |
| 70 | - | - | - | - | 2.95 | 0.95 | 3.17 | 1.02 |
| 75 | - | - | - | - | 2.81 | 0.91 | 3.26 | 1.05 |
| 80 | - | - | - | - | 2.67 | 0.86 | 3.13 | 1.01 |
| 85 | - | - | - | - | 2.54 | 0.82 | - | - |

We present the data in a chart to reveal trends and relationships (Figure 5).

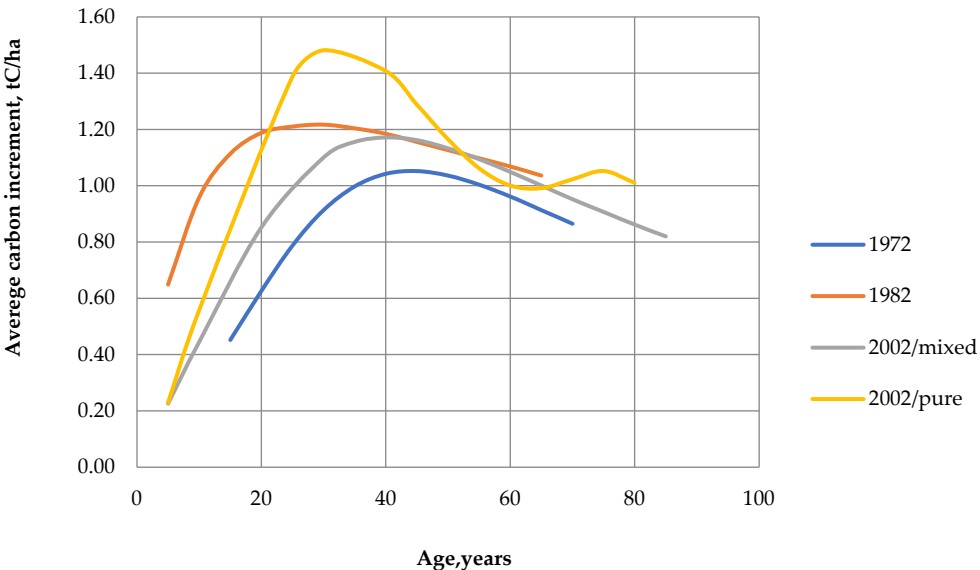

**Figure 5.** Trendlines indicating changes in the average annual carbon increment with age and inventory periods.

Starting from the age of 40, a decrease in the average carbon increment was observed in aspen forests. After 55 years, the average growth value leveled off, and the differences became insignificant (Figure 5).

Along with an increase in biomass growth rate in aspen stands, over the years, sustainability disturbances and loss of wood began to appear, due to natural and pathogenic mortality. Based on a yield table and analysis of the aspen stands inventory data, we defined the value of natural and pathogenic thinning of the growing stock, starting from the age of 50 years (Table 3).

**Table 3.** The values of natural and pathogenic loss of biomass and carbon in the aspen stands.

| Age, Years | M, m³/ha | Natural Mortality | | | | Pathogens-Induced Mortality | | | |
|---|---|---|---|---|---|---|---|---|---|
| | | % | Growing Stock, m³/ha | Phytomass, t/ha | Carbon, tc/ha | % | Growing Stock, m³/ha | Phytomass, t/ha | Carbon, tc/ha |
| 50 | 179 | 3 | 5.4 | 3.59 | 1.20 | 5 | 9.0 | 5.99 | 2.01 |
| 55 | 192 | 3 | 5.8 | 4.20 | 1.41 | 5 | 9.6 | 7.01 | 2.35 |
| 60 | 204 | 5 | 10.2 | 7.45 | 2.49 | 10 | 20.4 | 14.89 | 4.99 |
| 65 | 216 | 5 | 10.8 | 7.88 | 2.64 | 10 | 21.6 | 15.77 | 5.28 |
| 70 | 227 | 10 | 22.7 | 16.57 | 5.55 | 15 | 34.1 | 24.86 | 8.33 |
| 75 | 237 | 10 | 23.7 | 17.30 | 5.80 | 15 | 35.6 | 25.95 | 8.69 |
| 80 | 247 | 15 | 37.1 | 27.05 | 9.06 | 20 | 49.4 | 36.06 | 12.08 |

## 4. Discussion

Aspen has one of the highest carbon-sequestration potentials among all other forest-forming species in the boreal zone. In this sense, aspen stands are able to perform the same carbon sequestration as oak stands in temperate forests [16]. It was found that aspen forests are able to absorb up to 290 thousand tons of carbon per year in Krasnodar Krai (Russian Federation) [18,24,25].

The diagrams (Figure 2) show a higher average growth rate in 2002 than in 1972. However, there were significant differences in growth rate values between pure aspen stands (90%–100% of aspen in a stand composition) in relation to mixed ones. This is why we divided studied forest stands into two categories: pure and mixed.

The obtained data coincide with an age assessment of the average increment in modal aspen stands conducted by M.A. Danilin for the forest steppe zone of Central Siberia [11]. According to the scientist, the mixed aspen forests of this region (in the years 1950–1980) had an average growing stock increment of 3.2 m³/ha over 50 years, which indirectly confirms the positive current changes in the wood and carbon productivity of aspen forests. At the same time, in the Voronezh Oblast (Russian Federation), aspen plantations of the I *bonitet* class are capable of accumulating twice as much wood (6.6 m³/ha) and hence carbon, in 50 years [26]. Aspen stands of the Russian Far East at the northern limit of their range are capable of producing 1.7–2.9 m³/ha in 50 years [27].

Some researchers [28,29] believe that allometric equations can be used to create biomass maps for carbon assessment and forestry activities-planning. The resulting regressions of the average annual carbon increment over age have high correlation coefficients between the experimental and leveled data (R = 0.94–0.98), which allows their use when predicting the average increment.

Changes in the average stock increment are different for forest inventory periods. For example, there was a prominent upward trend in 1972, while in 1982, average stock increment values decreased, which can be explained by the absence of young stands (up to 30 years old). In 2002, both upward and downward trends were observed. With the start of aspen forests dieback in 2021, average stock increment values decreased (according to the data collected on the research plots, and analysis of the forest unit-level data for six forest compartments). Similar studies of the average increment dynamics were carried out in Tibet [30]. The authors stated that biomass carbon stocks first increased

(upward trend) and then decreased (downward trend) over the years, primarily due to tree growth characteristics.

A trend line (blue) shows an overall increase in the sum of temperatures from 1982 to 2020 (Figure 3). A red line shows that temperatures were below the median value from 1982 to 1995. There was a slight decline from 2001 to 2010, while the sum of temperatures still exceeded the median value. From 2011 to the present, there has been an intensive increase in the sum of temperatures (Figure 3).

The median amount of precipitation for the entire period (1982–2020) was 310.84 mm (Figure 4). The minimum value was set in 1989 (the amount of precipitation was 175.97 mm). The maximum value was recorded in 2020 and amounted to 428.23 mm. In 1982–2002, precipitation did not exceed the median value. From 2003 to the present, the amount of precipitation per season has been (insignificantly) growing (Figure 4).

There was a steady trend of increase in sum of temperatures during the growing season from 1982 to 2002 which corresponded with the period of intensive growth and development of the stands. However, the amount of precipitation at that time did not exceed the median value (Figures 3 and 4).

Based on the maximum trend line for the growth of pure aspen forests in 2002, we can state that with an increase in the sum of temperatures from 1982 to 2002 from 1800 °C to 2100 °C, the average carbon increment increased from 0.56 to 1.48 tons C/ha per year. This statement is true for pure aspen forests aged 10 to 30 years (Figure 5). The climatic factor (the sum of temperature during the growing season) does not have a direct impact on aspen forests' incremental characteristics. However, along with other abiotic and biotic factors, it creates favorable conditions for aspen growth. This is consistent with the conclusions made by Sergienko [13].

Natural mortality varied from 3 to 15% from 50 to 80 years of age. Pathogen-induced mortality for the same age period varied from 5 to 20%. For instance, according to Danilin, mature and over-mature aspen forests in Siberia are characterized by 50%–100% rot damage [10]. At the age of 80 years, thinning reached 30%, and the process of stands' dieback began. Carbon emission from 50 to 80 years of age increased from 1 to 12 tons C/ha. Some studies of aspen stands in mixed (coniferous–deciduous) forests of European Russia revealed a general pattern of decrease in average growth from 40 years for all forest types. The proportion of infected trees by the age of 50–60 reached 80% [31]. Thus, for all aspen forests in Russia, general age patterns are observed in terms of both average growth and state. Noteworthilt, the average growth rate depends on climate, which indirectly confirms our results.

Over forest development, the carbon stock increases both in the stand and in detritus [32]. It should be noted that, according to a number of scientists, aspen forests increase both the overall stability of forest ecosystems and the rhizospheric stability of carbon storage under global climate change [33].

## 5. Conclusions

This research allowed us to conclude that there was a steady increase in the growing season of temperature sum in the subtaiga/forest steppe region of Central Siberia (Russian Federation) from 1982 to 2002. The amount of precipitation at this time did not exceed the median value. The average growth rate of young pure aspen forests (up to 30 years old) depends on the sum of positive temperatures during the growing season. Nevertheless, stand composition (pure or mixed stands) influenced average growth rate more than climate change. We revealed a certain influence of climatic trends on aspen forests growth; however, this was indirect. From the age of 40, a decrease in the average carbon increment was observed in aspen stands. After 55 years, the average carbon increment values in different-composition aspen forests levels off, and the differences become insignificant. Between the ages of 50 and 80, aspen stands start losing sustainability and emission intensifies. At the age of 90, the mortality reaches 30%, and aspen stands begin to dieback.

Thus, it should be recommended to use young pure aspen forests to intensify carbon sequestration under climate change, by creating carbon farms, for example.

**Author Contributions:** Conceptualization, A.A.V. and V.N.N.; methodology, A.A.V., V.V.P. and A.A.A.; formal analysis, A.A.V., A.G.N. and V.N.N.; investigation, A.A.V., A.G.N. and V.N.N.; writing—original draft preparation, A.A.V.; writing—review and editing, A.A.V. and P.V.M.; visualization, A.G.N.; project administration, P.V.M.; funding acquisition, P.V.M. All authors have read and agreed to the published version of the manuscript.

**Funding:** The research was carried out within the framework of the state assignment established by the Ministry of Science and Higher Education of the Russian Federation for the implementation of the project "Assessment of the resilience of forest ecosystems to climate change as a basis for monitoring the carbon budget" (FEFE-2021-0018) by the team of the scientific laboratory "Forest Ecosystems".

**Data Availability Statement:** Not applicable.

**Conflicts of Interest:** The authors declare no conflict of interest.

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
