# Peer review of "Assessment of Carbon Productivity Dynamics in Aspen Stands under Climate Change Based on Forest Inventories in Central Siberia"

_forests, doi:10.3390/f14010109_

Round 1
Reviewer 1 Report
This is an interesting study of Aspen Forest in the subtaiga/forest-steppe region of Central Siberia. There is a high value on this study related to the extended period of valuable observations. My main criticism is related to a big part of the methodology is mixed in the Results section.
L. 18: “did not exceed the median value” of the historical precipitation? Of the precipitation between 1982 and 2021? Please clarify.
L.62: There is an extra “ю” at the end of the line.
L.64. Map coordinates in the graticule should be in English. Map on the right may show the position on the study area on the context of Russia.
L.68: The inventory years differ from the ones in the abstract.
L.72: Please enumerate all the indicators used and do not use “etc.”
L.102: m3/ha/year is incorrect. You can use m3 ha-1 year-1 or m3 / (ha year). Solidus/slash cannot be repeated on the same line.
L.112 remove plus sign.
L.136 to L.139 this is part of the methodology. The same for line 144, and L151-152.
L. 145: “we differentiated the initial data by five-year periods.” replace with “we differentiated the initial data by age in five-year increments.”
Author Response
Уважаемый коллега!
Большое Вам спасибо за Ваше время, усилия и советы, и особенно за рекомендации. Мы надеемся, что наши исправления вполне удовлетворительны.
Пожалуйста, ознакомьтесь с приложением.

Reviewer 2 Report
This manuscript investigates changes in carbon sequestration rate and productivity of aspen stands in Central Siberia using forest inventory. I have the following major concerns, which should be addressed to improve the clarity of the manuscript.
Major concerns:
1. The introduction is very simple. I strongly suggest rewriting this part. For example, a more critical literature review is required. What’s the limitation of previous studies? What’s the innovation of your study? What’s the objectives of your study? All these information and descriptions are very much needed.
2. The descriptions of the Methods are also relatively simple. More details regarding the processing methods and datasets should be provided. For example, what statistical methods were used in this study (Line 132)? The detailed information regarding the collected data (e.g., number of sites and collected variables) in each period is required (e.g., line 69-70). What are the data sources for climate data? And what is the spatial resolution?
3. An in-depth discussion of the results is needed. The current discussion is relatively simple. I suggest that the authors compare their estimations with previous studies and include more mechanistic explanations.
Author Response
Dear colleague, we're grateful to You for Your time, comments and advices.
We really appreciate the informational help that will help improve our manuscript.
Please see the attachment.

Round 2
Reviewer 2 Report
I don't think my concerns have been well addressed.
Author Response
Dear colleague, we're grateful to You for Your time, comments and advices.
We really appreciate the informational help that will help improve our manuscript.
Major concerns:
- The introduction is very simple. I strongly suggest rewriting this part. For example, a more critical literature review is required. What’s the limitation of previous studies? What’s the innovation of your study? What’s the objectives of your study? All these information and descriptions are very much needed.
Authors':
Despite the large number of studies focused on carbon regulation in Russia, the task of creating a database of carbon content in ecosystems has not lost its relevance. Notably, studies of long-term changes in deciduous species in Siberia were carried out very limitedly and fragmentarily on individual research plots. The purpose of the present study was to assess the productivity of aspen forests by the average increments of woody biomass and carbon in it, considering the stand composition, age and climate changes.
The introduction has been completely rewritten. Please take a look.
- The descriptions of the Methods are also relatively simple. More details regarding the processing methods and datasets should be provided. For example, what statistical methods were used in this study (Line 132)? The detailed information regarding the collected data (e.g., number of sites and collected variables) in each period is required (e.g., line 69-70). What are the data sources for climate data? And what is the spatial resolution?
Authors':
Information about processing methods and data sets has been added to the methodology section.
Statistical processing in L.128-133 and 147-148.
The number of sites and the collected variables are reflected in the lines 91-99.
The data sources for climate data and their spatial resolution are given in the lines 152-156.
- An in-depth discussion of the results is needed. The current discussion is relatively simple. I suggest that the authors compare their estimations with previous studies and include more mechanistic explanations.
Authors':
The Discussions has been completely rewritten. Please take a look

Round 3
Reviewer 2 Report
I have no more concerns